# Incidence Trend of Follicular Lymphoma in Taiwan Compared to Japan and Korea, 2001–2019

**DOI:** 10.3390/jcm12041417

**Published:** 2023-02-10

**Authors:** Liang-Chun Chiu, Chih-Wen Lin, Hung-Ju Li, Jian-Han Chen, Fu-Cheng Chuang, Sheng-Fung Lin, Yu Chang, Yu-Chieh Su

**Affiliations:** 1Department of Obstetrics and Gynecology, E-Da Dachang Hospital, I-Shou University, Kaohsiung 807, Taiwan; 2School of Medicine, College of Medicine, I-Shou University, Kaohsiung 824, Taiwan; 3Division of Gastroenterology and Hepatology, Department of Medicine, E-Da Hospital, I-Shou University, Kaohsiung 824, Taiwan; 4Division of Hematology-Oncology, Department of Internal Medicine, E-Da Hospital, I-Shou University, Kaohsiung 824, Taiwan; 5Bariatric and Metabolic International Surgery Center, E-Da Hospital, I-Shou University, Kaohsiung 824, Taiwan; 6Division of General Surgery, Department of Surgery, E-Da Hospital, I-Shou University, Kaohsiung 824, Taiwan; 7Department of Radiation Oncology, E-Da Hospital, I-Shou University, Kaohsiung 824, Taiwan; 8Department of Obstetrics and Gynecology, E-Da Hospital, I-Shou University, Kaohsiung 824, Taiwan; 9School of Medicine for International Students, College of Medicine, I-Shou University, Kaohsiung 824, Taiwan

**Keywords:** follicular lymphoma, incidence trend, Taiwan, Japan, South Korea

## Abstract

A continuous increase in follicular lymphoma has been observed in Taiwan, Japan, and South Korea over the last few decades. This study aimed to evaluate the difference in incidence trends of follicular lymphoma in Taiwan, Japan, and South Korea between 2001 and 2019. The data for the Taiwanese populations was obtained from the Taiwan Cancer Registry Database, and those for the Japanese and Korean population were retrieved from the Japan National Cancer Registry and some additional reports, both of which included population-based cancer registry data, from Japan and Korea. Follicular lymphoma accounted for 4231 cases from 2002–2019 in Taiwan, 3744 cases from 2001–2008 and 49,731 cases from 2014–2019 in Japan; and 1365 cases from 2001–2012 and 1244 cases from 2011–2016 in South Korea. The annual percentage change for each time period was 3.49% (95% confidence interval: 2.75–4.24%) in Taiwan, 12.66% (95% confidence interval [CI]: 9.59–15.81%) and 4.95% (95% CI: 2.14–7.84%) in Japan, and 5.72% (95% CI: 2.79–8.73%) and 7.93% (95% CI: −1.63–18.42%) in South Korea. Our study confirms that the increasing trends of follicular lymphoma incidence in Taiwan and Japan have been remarkable in recent years, especially the rapid increase in Japan between 2014 and 2019; however, there was no significant in-crease from 2011 to 2015 in South Korea.

## 1. Introduction

Non-Hodgkin’s lymphoma (NHL) is the most common hematological malignancy worldwide, with an increasing incidence over the last few decades [1,2,3]. NHL subtypes have varying clinical behaviors and biological heterogeneity, and the patterns of incidence and time trend of each subtype, sex, age, race/ethnicity, and geographic region are evaluated [4,5,6,7].

The literature has listed possible risk factors for follicular lymphoma (FL): a family history of NHL (1.99-fold increased risk of FL), hepatitis C virus (a 2.73-fold increased risk of FL), select autoimmune diseases, allergy/atopy, and alcohol consumption [8,9]. The incidence of FL has shown a rising trend in some areas, such as Canada, Poland, Japan, South Korea, and Taiwan; however, the reasons for this remain unknown [2,10,11,12,13,14,15,16,17].

Only a few studies have investigated the incidence of FL in Asia owing to its relative rarity in Asia compared with western countries [7,12,13,14,15,16,17,18]. In Singapore, the time trend in the incidence of NHL among men and women increased in 1998–2012, but the incidence of FL did not significantly increase [19]. Su et al. demonstrated that the incidence of FL in Taiwan had an increasing trend and varied among the other ethnic groups in the United States between 2008 and 2017 [16]. The incidence in Japan from 1993 to 2014 and in the Republic of Korea from 1999 to 2015 also showed an upward trend [2,12,13,14]. In Sweden, the prevalence of FL increased in 2004–2016, but the incidence remained unchanged [1]. In Canada, the overall incidence of FL increased from 1992 to 2010 [10].

This study aimed to evaluate the differences in incidence and time trends of FL in Taiwan, Japan, and South Korea from population-based cancer registries because of the difficulty in obtaining detailed data for other countries in Asia.

## 2. Materials and Methods

In this study, data were obtained from institutions and previous studies. The number of FL cases in Taiwan between 2008–2019 was obtained from the Cancer Registry Annual Report of the Health Promotion Administration, Ministry of Health and Welfare, Taiwan [20]. The annual population size for calculating incidence was obtained from the Department of Household Registration, Ministry of the Interior, Taiwan [21]. These annual reports were based on the Taiwan Cancer Registry Database (TCRD), a high-quality database from a nationwide population-based cancer registry system. In terms of the quality indicators of the registry, the completeness rates ranged from 91.3% in 2001 to 98.3% in 2019, the percentage of death certificate–only cases in all cancer cases (DCO%) decreased from 2.6% in 2003 to 0.7% in 2019, while the percentage of morphological verification (MV%) increased from 87.1% in 2003 to 93.5% in 2019 [15,22,23].

Cases of patients with FL were identified using the International Classification of Disease for Oncology, Third Edition (ICD-O-3; Histology codes: 9690, 9691, 9695, and 9698) [20,24].

Age-standardized incidence rates (ASRs) and male-to-female incidence rate ratios (M/F IRRs) were calculated. ASR was age-adjusted to the 2000 world standard population as defined by the World Health Organization [25]. The temporal trends for the incidence of both sexes were described using the annual percent change (APC) calculated using the Joinpoint Regression Program, Version 4.9.1.0 (NCI Statistical Methodology and Applications Branch, Bethesda, MD, USA) [26]. The APCs were estimated by observing the changes in the trend on a log scale and assuming constant variance.

We further combined the data, based on the TCRD, in Taiwan between 2002–2007 as reported by Ko et al. [15] and compared the ASRs with those reported in Japan and South Korea. Japanese data were retrieved from Chihara et al. and the National Cancer Center, Japan, both of which are population-based cancer registry data [12,27]. Korean data were obtained from Lee et al. and Kim et al.; the data were from the Korea Central Cancer Registry, (a hospital-based nationwide cancer registry) and from the National Health Information Database, (a public database that covered the entire Korean population), respectively [2,13]. Based on the ASRs provided in these reports, we computed the APCs for the Japanese and Korean population.

An independent two-sided t-test was used to determine whether APC was statistically significant from zero. A *p*-value < 0.05 was considered statistically significant. Incidence rate ratios (IRRs) were calculated using the numbers of FL cases for each area and the age-specific population structure for each year. Statistical significance in difference between IRRs was determined whether the 95% confidence intervals (CIs) overlap between IRRs or not.

## 3. Results

A total of 4231 cases of FL in Taiwan were recorded between 2002 and 2019, including 2181 (51.5%) men and 2050 (48.5%) women. The data for Japan included 3744 cases (men, *n* = 1870 [49.9%]; women, *n* = 1874 [50.1%]) in 2001–2008 and 49,731 cases (men, *n* = 23,459 [47.2%]; women, *n* = 26,271 [52.8%]) in 2014–2019. South Korea recorded 1365 cases in 2001–2012 and 1244 cases in 2011–2015. Table 1 shows the ASRs (per 100,000) and male-to-female IRR for FL in Taiwan and Japan.

Comparisons of incidence trends of FL among Taiwan, Japan, and South Korea are depicted in Figure 1, Figure 2 and Figure 3, and detailed data are described in Appendix A. Overall ASR per 100,000 increased significantly: in Taiwan, from 0.48 in 2002 to 1.07 in 2019, representing a 123% increase and an APC of 3.49% (95% CI: 2.75–4.24%); in Japan, from 0.43 in 2001 to 1.08 in 2008 (APC: 12.66%, 95% CI: 9.59–15.81%), representing a 150% increase, and from 2.54 in 2014 to 3.17 in 2019 with an APC of 4.95% (95% CI: 2.14–7.84%) representing a 25% increase; and in South Korea, approximately two times larger in 2012 (0.28/100,000) than that in 2001 (0.14/100,000), with an APC of 5.72% (95% CI: 2.79–8.73%), whereas no significantly increase was seen from 2011 (0.50/100,000) to 2015 (0.63/100,000) with an APC of 7.93% (95% CI: −1.63–18.42%). ASR among men in Taiwan increased with an APC of 2.60% (95% CI: 1.74–3.46%) from 2002 to 2019, and that in Japan continuously rose until 2019 (2001–2008: APC = 12.60%, 95% CI: 7.12–18.36%; 2014–2019: APC = 5.02%, 95% CI: 1.93–8.21%) (Figure 2). ASR among women in Taiwan significantly increased between 2002 and 2019 (APC = 4.54%, 95% CI: 3.59–5.50%), and the APC of FL for Japanese women showed three continuous upward trends (2001–2005: APC = 8.86%, 95% CI: 5.53–12.30%; 2005–2008: APC = 19.14%, 95% CI: 13.43–25.14%; AAPC in 2001–2008 = 13.15%, 95% CI: 11.25–15.09%. 2014–2019: APC = 4.88%, 95% CI: 2.02–7.81%) (Figure 3).

The male-to-female IRRs in Taiwan and Japan are described in Table 1. The IRRs for Taiwanese and Japanese ranged from 0.89 (95% CI: 0.69–1.16) to 1.51 (95% CI: 1.12–2.04), and from 0.80 (95% CI: 0.67–0.95) to 1.28 (95% CI: 1.07–1.53), respectively.

Table 2 shows the Japan-to-Taiwan IRRs by sex for 2002, 2008, and 2019. FL were three times as common in the Japanese compared with the Taiwanese from 2014 to 2019, ranging from an IRR of 3.22 (95% CI: 2.90–3.57) to 5.06 (95% CI: 4.51–5.69) (Appendix A). The IRR significantly increased between 2002 and 2019 for overall FL cases, which increased from 1.14 (95% CI: 0.92–1.41) in 2002 to 2.95 (95% CI: 2.67–3.26) in 2019. The same trends were also shown for both sexes (men: 1.04 in 2002 to 2.90 in 2019; women: 1.22 in 2002 to 3.02 in 2019). Table 3 shows the Taiwan-to-Korea IRRs for overall FL cases in 2002 and 2012, 2011, and 2015. IRR did not significantly change in 2002–2012 and 2011–2015 and remained between 2.74 (95% CI: 2.27–3.30) and 3.00 (95% CI: 3.28–3.95) in 2002–2012 and between 1.52 (95% CI: 1.26–1.84) and 1.56 (95% CI: 1.34–1.82) in 2011–2015.

## 4. Discussion

The incidence of FL has increased in some countries but decreased for Whites in the United States [2,10,11,12,13,14,15,16,17]. Although Su et al. showed that the incidence trends did not change significantly for male and female Asians and Pacific Islanders living in the United States, our study suggested that the Japanese population had a rapid increase in FL incidence and a moderately increasing trend was observed in the Taiwanese population [16]. In Korea, an upward trend was found from 2001 to 2015, but no significant change was reported in the 2011–2015 period. This result is consistent with a report from a large tertiary institution in South Korea (Asan Medical Center, Seoul, Republic of Korea) which showed that the age-adjusted incidence of FL significantly increased between 2008 and 2017 [14].

It is worth mentioning that the upward trend in B-cell lymphoma incidence in Taiwan also occurred during the period 2001–2019; moreover, the relative frequency of FL among B-cell lymphoma cases increased from 2008 to 2019, with an APC of 1.47% (95% CI: 0.10–2.86%). This indicates that the FL incidence grew more quickly than that of other types of B-cell lymphoma in Taiwan. In Korea, the incidence rate of B-cell lymphoma also increased from 3.71 in 2001 to 6.60 in 2012 (obtained from Lee et al. [13]), and from 5.74 in 2011 to 6.96 in 2015 [2]. Nevertheless, there was no significant change in the relative frequency of FL among B-cell lymphoma cases during the study period. In Japan, the relative frequency of FL among NHL rose from 0.11 in 2001 to 0.15 in 2008, according to Chihara et al. [12], and from 0.24 in 2014 to 0.27 in 2019, according to the database of the National Cancer Registry in Japan [27]). It shows that among Japanese people, FL incidence increased more swiftly than other subtypes of NHL between 2001 and 2019.

Although the APC was higher among women than among men in Taiwan, the incidence was higher among men than among women in most years. This result is consistent with results reported by Su et al.: the APCs for men and women were −0.3 and 1.7, respectively, but the incidence rates were higher among men than among women every year from 2008 to 2017 [16]. On the contrary, the male-to-female IRR exceeded 1 in most years from 2001 to 2008 in Japan, but due to the sharp increase in FL incidence among Japanese women between 2014 and 2019, the IRRs ranged from 0.93 to 0.99. Among these three countries, there has been a marked difference between the incidence rates of FL in Japan and Taiwan since 2008. By 2019, the incidence rate for the Japanese was 195% higher than for the Taiwanese (Table 2). Compared with Koreans, the FL incidence rate was about 50% higher among the Taiwanese (Table 3).

The reasons for the continuous increase in FL in Asia are still unclear. Several studies examining risk factors for FL provided evidence that patients engage in behaviors including smoking, a sedentary lifestyle, poor diet, environmental exposure, recreational sun exposure, and obesity [8,9,28,29]. Obesity has a significant and positively strong association with cancer incidence, manifested in patients with other malignancies, and has become a potential risk factor for FL, which may explain the upward trend on FL in Taiwan and South Korea [9,30]. A survey on the change of national nutrition and health status in Taiwan indicated that the prevalence of obesity (BMI ≥ 24 kg/m^2^) has been increasing from 51.6% (2005–2008) to 58.9% (2017–2020) and from 36.4% (2005–2008) to 42.8% (2017–2020) for men and women in Taiwan, respectively [31]; in South Korea, the prevalence of class I (BMI 30–34.9), class II (BMI 35–39.9), and class III (BMI ≥ 40) obesity increased significantly from 29.1%, 3.2%, and 0.3% in 2009 to 32.5%, 5.2%, and 0.8% in 2018, respectively. However, the proportion of obesity (BMI ≥ 25) in Japan ranged from 28.6% to 32.2% in men and from 19.2% to 21.9% in women, with no significant change between 2008 and 2018 in both sexes [32]. The increased identification of FL might arise from advancements in diagnostic methods and precision (early detection of cancer is also a result of screening procedures, technological improvements, etc.).

Statistical evidence for an inverse association between FL and recreational sunlight exposure has been found [9,33]. Some studies in Taiwan found that vitamin D deficiency was partly due to a lack of sun exposure [34,35], similar to those in Japan, suggesting the total amount of sun exposure as a factor [36]. One study found that vitamin D deficiency is a very common health problem in South Korea, and that one possible cause is sunshine duration [37]. Further studies are needed to confirm that the increased incidence of FL in Taiwan, Japan, and South Korea is related to sun exposure.

Our study has some limitations. First, the FL data in Japan between 2001 and 2008 and those in Korea were not the primary data from the databases, but were retrieved from D. Chihara et al., Lee et al., and Kim et al. [2,12,13]. As a result, we cannot estimate the APCs by age group. Second, the standard populations used to age-adjust the incidence rates for these three areas were different, and a directed comparison should be made with caution. The incidences for Taiwan were age-adjusted to the 2000 world standard population as defined by the World Health Organization, those for Japan and Korea in 2001–2012 were age-adjusted to the Segi’s world standard population, and those for Korea in 2011–2015 were age-adjusted to the Korean population in 2011. In addition, even though we found that the median age of FL in Taiwan had risen from 55–59 years to 60–64 years (since 2016), the median age of FL patients in Japan and South Korea cannot be determined from the available data sources.

## 5. Conclusions

Our study confirms that the incidence trends of FL in Taiwan, Japan, and South Korea have been increasing in recent years, especially in Japan from 2014 to 2019. Based on sex, in Taiwan, women are more prone to FL than men. The APC of FL in Japan was higher among women than men between 2001 and 2008; however, the inverse was true from 2014 to 2019. During this study period, the male-to-female IRRs in Taiwan and Japan did not show a significant change. Further investigations regarding the risk factors of the increasing trends of FL incidence in Asian countries are warranted.

## Figures and Tables

**Figure 1 jcm-12-01417-f001:**
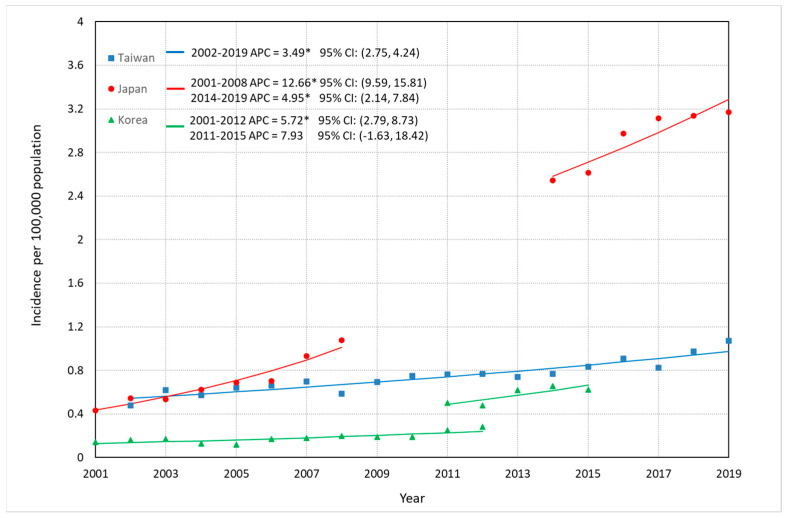
Trends in the ASRs of FL in Taiwan, Japan and Korea. * The APC is significantly different from 0 at the alpha level of 0.05.

**Figure 2 jcm-12-01417-f002:**
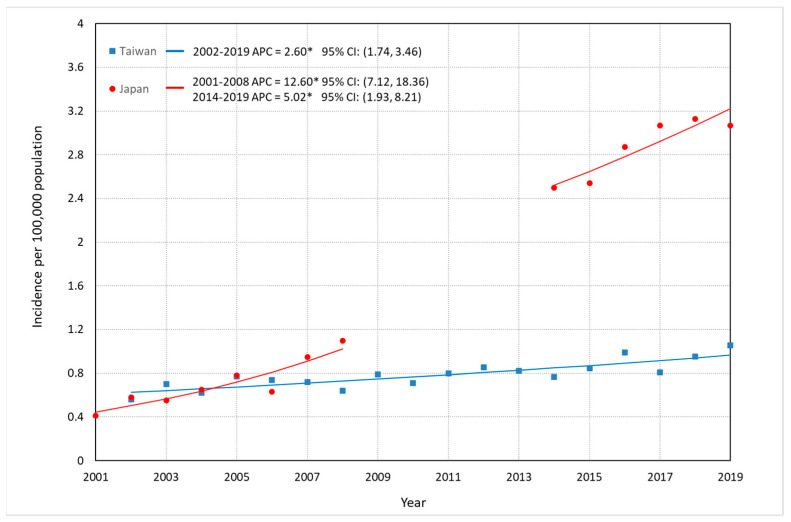
Trends in the ASRs of FL for males in Taiwan and Japan. * The APC is significantly different from 0 at the alpha level of 0.05.

**Figure 3 jcm-12-01417-f003:**
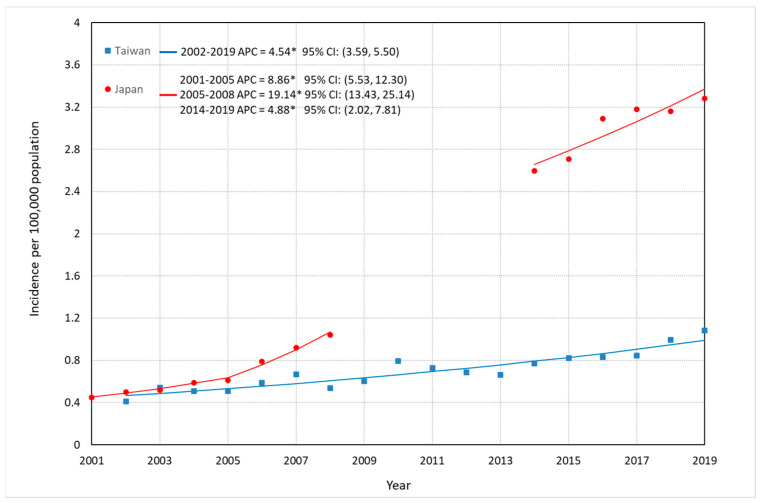
Trends in the ASRs of FL for females in Taiwan and Japan. * The APC is significantly different from 0 at the alpha level of 0.05.

**Table 1 jcm-12-01417-t001:** Age-adjusted rates per 100,000 and male-to-female IRR in Taiwan and Japan.

		Taiwan				Japan		
Year	Overall	Male	Female	IRR ^1^ (95% CI ^2^)	Overall	Male	Female	IRR ^1^ (95% CI ^2^)
2001					0.43	0.41	0.45	0.91 (0.70, 1.18)
2002	0.48	0.56	0.41	1.37 (0.95, 1.97)	0.55	0.58	0.50	1.16 (0.92, 1.46)
2003	0.62	0.70	0.54	1.30 (0.95, 1.78)	0.54	0.55	0.52	1.06 (0.86, 1.30)
2004	0.57	0.62	0.51	1.22 (0.88, 1.69)	0.63	0.65	0.59	1.10 (0.91, 1.33)
2005	0.64	0.77	0.51	1.51 (1.12, 2.04)	0.69	0.78	0.61	1.28 (1.07, 1.53)
2006	0.66	0.74	0.59	1.25 (0.94, 1.68)	0.71	0.63	0.79	0.80 (0.67, 0.95)
2007	0.70	0.72	0.67	1.07 (0.81, 1.43)	0.93	0.95	0.92	1.03 (0.89, 1.20)
2008	0.59	0.64	0.54	1.19 (0.88, 1.62)	1.08	1.10	1.04	1.06 (0.92, 1.22)
2009	0.69	0.79	0.60	1.31 (1.00, 1.73)	--	--	--	--
2010	0.75	0.71	0.79	0.89 (0.69, 1.16)	--	--	--	--
2011	0.76	0.80	0.73	1.10 (0.85, 1.42)	--	--	--	--
2012	0.77	0.85	0.69	1.24 (0.97, 1.59)	--	--	--	--
2013	0.74	0.82	0.66	1.24 (0.96, 1.60)	--	--	--	--
2014	0.77	0.77	0.77	0.99 (0.78, 1.26)	2.54	2.50	2.60	0.96 (0.92, 1.01)
2015	0.83	0.85	0.82	1.03 (0.82, 1.30)	2.62	2.54	2.71	0.94 (0.89, 0.98)
2016	0.91	0.99	0.83	1.19 (0.95, 1.48)	2.97	2.87	3.09	0.93 (0.89, 0.97)
2017	0.82	0.81	0.85	0.96 (0.76, 1.20)	3.12	3.07	3.18	0.96 (0.93, 1.01)
2018	0.98	0.95	0.99	0.96 (0.78, 1.18)	3.14	3.13	3.16	0.99 (0.95, 1.03)
2019	1.07	1.06	1.09	0.97 (0.80, 1.18)	3.17	3.07	3.28	0.93 (0.90, 0.97)

^1^ Male-to-female incidence rate ratio. ^2^ Confidence interval.

**Table 2 jcm-12-01417-t002:** Japan-to-Taiwan incidence rate ratios by gender, 2002, 2008, and 2019.

Year	IRR ^1^ (95% CI ^2^) Overall	IRR (95% CI) Male	IRR (95% CI) Female
2002	1.14 (0.92, 1.41)	1.04 (0.78, 1.38)	1.22 (0.88, 1.69)
2008	1.83 (1.55, 2.16)	1.72 (1.37, 2.16)	1.94 (1.52, 2.47)
2019	2.95 (2.67, 3.26)	2.90 (2.50, 3.35)	3.02 (2.64, 3.46)

^1^ Incidence rate ratio. ^2^ Confidence interval.

**Table 3 jcm-12-01417-t003:** Taiwan-to-Korea incidence rate ratios.

Year	IRR ^1^ (95% CI ^2^)
Korea_Lee ^3^	
2002	3.00 (2.28, 3.95)
2012	2.74 (2.27, 3.30)
Korea_Kim ^4^	
2011	1.52 (1.26, 1.84)
2015	1.56 (1.34, 1.82)

^1^ Incidence rate ratio. ^2^ Confidence interval. ^3^ Data were obtained from Lee et al. [13]. ^4^ Data were obtained from Kim et al. [2].

## Data Availability

All relevant data are within the manuscript and its Supporting Information files.

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
