# Peer review of "Incidence Trend of Follicular Lymphoma in Taiwan Compared to Japan and Korea, 2001–2019"

_jcm, 2023, doi:10.3390/jcm12041417_

Round 1
Reviewer 1 Report
Has the incidence rate of other common types of B-cell lymphoma also increased between 2001-2019? Perhaps, the increased incidence is not restricted to FL but rather represents an overall increase in the risk of lymphoma. Did the authors notice any change in the median age at presentation of FL during this time period?
Author Response
Point 1: Has the incidence rate of other common types of B-cell lymphoma also increased between 2001-2019? Perhaps, the increased incidence is not restricted to FL but rather represents an overall increase in the risk of lymphoma.
Response 1: Thank you for your comments. We have provided it in the second paragraph in the “Discussion” chapter and marked it in red (page 6):
It is worth mentioning that the upward trend in B-cell lymphoma incidence in Taiwan also occurred during the period 2001–2019, moreover, the relative frequency of FL among B-cell lymphoma cases increased from 2008 to 2019, with an APC of 1.47% (95% CI: 0.10%–2.86%). It indicates that FL incidence grew more quickly than that of other types of B-cell lymphoma for Taiwanese. In Korea, the incidence rate of B-cell lymphoma also increased from 3.71 in 2001 to 6.60 in 2012 (obtained from Lee et al. [13]), and from 5.74 in 2011 to 6.96 in 2015 [2]. Nevertheless, there was no significantly change in the relative frequency of FL among B-cell lymphoma cases during the study period. In Japan, the relative frequency of FL among NHL rose from 0.11 in 2001 to 0.15 in 2008, according to Chihara et al. [12], and from 0.24 in 2014 to 0.27 in 2019, according to the database from National Cancer Registry in Japan [27]). It shows that among Japanese people, FL incidence increased more swiftly than other subtypes of NHL between 2001 and 2019.
Point 2: Did the authors notice any change in the median age at presentation of FL during this time period?
Response 2: Thank you for your comments. We have provided it in the last paragraph of the Discussion chapter and marked it in red (page 7):
In addition, even though we found the median age of FL in Taiwan has risen from 55-59 years to 60-64 years (since 2016), but the median age of FL patients in Japan and South Korea cannot be determined from the available data sources.

Reviewer 2 Report
Dear Authors, the present epidemiological study is an interesting addition to the current knowledge of follicular lymphoma incidence and prevalence in three asian Countries: Japan, Taiwan and South Korea.
Here you find some issues that need to be addressed:
1. Discussion chapter: please, comment the significance and the utility of the IRR parameter . It can be unclear to some of the readers (see the results in Table 2 and Table 3).
2. In the "Discussion", I think that the improvements made in the diagnostics approaches and accuracy may lead also to a higher number of diagnosed follicular lymphomas (early detection of cancer is also a result of screening procedures, technological improvements, etc). This point can be mentioned among the possible causes for the observed FL increase in the selected Countries.
Author Response
Point 1: Discussion chapter: please, comment the significance and the utility of the IRR parameter. It can be unclear to some of the readers (see the results in Table 2 and Table 3).
Response 1: Thank you for your reminder. We have provided it in the third paragraph of the Discussion chapter and marked it in red (page 6):
Among these three countries, there has been a marked difference between the incidence rates of FL in Japan and Taiwan since 2008. By 2019, the incidence rate for Japanese was 195% higher than for Taiwanese (Table 2). Compare with Korean, FL incidence rate was about 50% higher among Taiwanese (Table 3).
Point 2: In the "Discussion", I think that the improvements made in the diagnostics approaches and accuracy may lead also to a higher number of diagnosed follicular lymphomas (early detection of cancer is also a result of screening procedures, technological improvements, etc). This point can be mentioned among the possible causes for the observed FL increase in the selected Countries.
Response 2: Thank you for your comments. We have mentioned it in the fourth paragraph and marked it in red (page 6):
Increased identification of FL might arise from advancements in diagnostic methods and precision (early detection of cancer is also a result of screening procedures, technological improvements, etc).

Round 2
Reviewer 1 Report
The authors have responded adequately. I have no further comments.